# ANY-SCALE BALANCED SAMPLERS FOR DISCRETE SPACES

**Haoran Sun**[*†]
hsun349@gatech.edu

**Bo Dai**[‡†]
bodai@google.com

**Charles Sutton**[‡]
charlessutton@google.com

**Dale Schuurmans** [‡§]
schuurmans@google.com

**Hanjun Dai**[‡]
hadai@google.com

## ABSTRACT

The locally balanced informed proposal has proved to be highly effective for sampling from discrete spaces. However, its success relies on the "local" factor, which ensures that whenever the proposal distribution is restricted to be near the current state, the locally balanced weight functions are asymptotically optimal and the gradient approximations are accurate. In seeking a more efficient sampling algorithm, many recent works have considered increasing the scale of the proposal distributions, but this causes the "local" factor to no longer hold. Instead, we propose *any-scale balanced samplers* to repair the gap in non-local proposals. In particular, we substitute the locally balanced function with an any-scale balanced function that can self-adjust to achieve better efficiency for proposal distributions at any scale. We also use quadratic approximations to capture curvature of the target distribution and reduce the error in the gradient approximation, while employing a Gaussian integral trick with a special estimated diagonal to efficiently sample from the quadratic proposal distribution. On various synthetic and real distributions, the proposed sampler substantially outperforms existing approaches.

## 1 INTRODUCTION

The Markov Chain Monte Carlo (MCMC) algorithm is one of the most widely used methods for sampling from intractable distributions (Robert et al., 1999). Gradient-based samplers that leverage gradient information to guide the proposal have achieved significant advances in sampling from continuous spaces, demonstrated, for example, by the Metropolis Adjusted Langevin Algorithm (MALA) (Rossky et al., 1978), Hamiltonian Monte Carlo (HMC) (Duane et al., 1987), and related variants (Girolami & Calderhead, 2011; Hoffman et al., 2014). However, for discrete spaces, gradient based samplers remain far less well understood. Recently, a family of locally balanced (LB) samplers (Zanella, 2020; Grathwohl et al., 2021; Sun et al., 2021; 2022a; Zhang et al., 2022) have demonstrated promise in sampling from discrete spaces. Such samplers use a locally balanced weight function in an informed proposal $Q(x, y) \propto g(\pi(y)/\pi(x))K_\sigma(x - y)$, such that $g : \mathbb{R} \to \mathbb{R}$ is a weight function that satisfies $g(t) = tg(\frac{1}{t})$, $\pi$ is the target distribution, and $K_\sigma$ is a kernel that determines the scale of the proposal distribution. It is also shown that such a locally balanced informed proposal is a discrete version of MALA, since they both simulate gradient flows in the Wasserstein manifold (Sun et al., 2022a).

In initial work, Zanella (2020) considered a local proposal with a kernel $K_\sigma$ that restricts next states to lie within a 1-Hamming ball, seeking to capture natural discrete topological structure arising, for example, in spaces of trees, partitions or permutations. For more regular discrete spaces, such as lattices, Grathwohl et al. (2021) introduce a gradient approximation for the probability ratio $\pi(y)/\pi(x) \approx \exp(\langle y - x, \nabla \log \pi(x) \rangle)$ to make the locally balanced proposal more scalable.

---

[*]Work done during an internship at Google.

[†]Georgia Tech

[‡]Google Research, Brain Team

[§]University of Alberta

However, by restricting attention to a local proposal, these methods tend not to make large jumps and exhibit highly correlated samples. Sun et al. (2021) made the first provably efficient attempt to extend local proposals from 1-Hamming ball to $L$-Hamming ball, after which subsequent works (Zhang et al., 2022; Sun et al., 2022a; Rhodes & Gutmann, 2022) have shown that using a non-local proposal for the heat kernel $K_\sigma(z) = \exp(-\frac{1}{2\sigma}\|z\|^2)$ can further improve sampling efficiency.

Even though extending locally balanced samplers to non-local proposals has delivered some progress, there remain opportunities for improvement by closing gaps in the current methods. One gap is exemplified by the choice of weight function. To illustrate, consider $g(t) = t^\alpha$. For a 100 dimensional Bernoulli distribution, we used an informed proposal with the heat kernel $K_\sigma(z) = \exp(-\frac{1}{2\sigma}\|z\|^2)$ and plotted the effective sample size as a function of $\alpha$ for different $\sigma$. Figure 1 shows clearly that performance of $\alpha$ varies for different $\sigma$. In particular, the optimal choice of $\alpha$ monotonically increases with $\sigma$. When $\sigma \downarrow 0$, the optimal choice $g(t) = \sqrt{t}$ recovers the locally balanced function. This result indicates that the locally balanced function is no longer optimal for non-local proposals. We will show that a good choice of $\alpha$ depends on the variance ratio between the target distribution and the kernel. We also give an adaptive algorithm that tunes $(\sigma, \alpha)$ automatically.

Another gap arises from the gradient approximation. For the local proposal, a first order gradient approximation is usually sufficient to estimate the probability ratio. However, for a non-local proposal, higher order approximations are generally required to capture correlations between different variables. Extending from recent work, we consider a quadratic approximation of the probability ratio: $\pi(y)/\pi(x) = \exp((y-x)^\top \nabla \log \pi(x) + \frac{1}{2}(y-x)^\top W(y-x))$ for non-local proposal, where $W$ is an arbitrary symmetric real matrix. Unfortunately, the

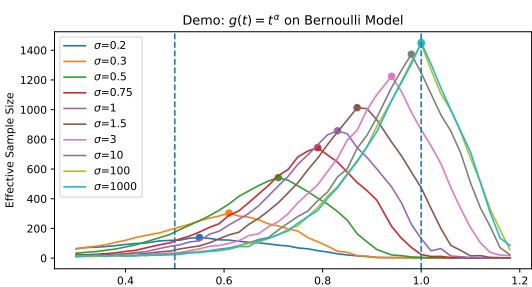

Figure 1: ESS for different $(\sigma, \alpha)$ pairs

quadratic heat kernel renders a proposal distribution that is a pairwise Markov random field in general, which is intractable to directly sample from. However, this difficulty can be addressed by leveraging a stochastic factorization via the Gaussian integral trick (Hertz et al., 1991; Zhang et al., 2012), also known as the Hubbard-Stratonovich transform (Hubbard, 1959). In particular, we decompose the quadratic term via $(W + D)^{\frac{1}{2}}\xi$, where $D$ is a diagonal matrix to make sure $W + D$ is positive semi-definite (PSD) and $\xi$ is standard Gaussian noise. In this paper we will show that the quality of the factorization can be characterized by $D$. While previous work chose $D$ to be isotropic, we find a substantial increase in performance by numerically optimizing over general diagonal matrices $D$.

Closing these two gaps renders our proposal for *Any-scale Balanced Sampling (AB Sampling)* methods. We extensively demonstrate the advantages of the proposed sampler on both synthetic and real distributions. The results show that, with the proposed numerical optimization of $D$ and the adaptive tuning, the two extensions robustly improve the efficiency of non-local informed proposal.

## 2 PRELIMINARIES

**Informed Proposal**. The informed proposal (Zanella, 2020) is a class of Metropolis-Hastings algorithms for discrete spaces, such that the proposal distribution at the current state $x$ has the form:

$$Q_\sigma^g(x, y) = g\left(\frac{\pi(y)}{\pi(x)}\right) K_\sigma(x - y)/Z_g(x), \quad Z(x) = \sum_{z \in \mathcal{X}} g\left(\frac{\pi(z)}{\pi(x)}\right) K_\sigma(x - y), \quad (1)$$

where $\pi$ is the target distribution, $\mathcal{X}$ is the state space, $g : \mathbb{R}_+ \to \mathbb{R}_+$ is a weight function, $Z$ is the partition function, and $K_\sigma$ is an uninformed kernel with size $\sigma$, such that $K(z) = K(-z)$, $\lim_{\sigma \to 0} K_\sigma(z) = 1_{\{z=0\}}$ and $\lim_{\sigma \to \infty} K_\sigma(z) \equiv 1$. For example, $K_\sigma$ can be a Hamming ball kernel $K_\sigma(z) = 1_{\{|z| \le \sigma\}}$, or a heat kernel $K_\sigma(z) = \exp(-\|z\|^2/2\sigma)$.

**Balanced Proposal**. Let $\pi_\sigma^g$ denote the reversible distribution associated with the informed proposal $Q_\sigma^g$; that is, satisfying $\pi_\sigma^g(x)Q_\sigma^g(x, y) = \pi_\sigma^g(y)Q_\sigma^g(y, x)$. We refer to the family $\{Q_\sigma^g\}_\sigma$ as a balanced proposal if there exists a sequence $\sigma_1, \sigma_2, ...,$ such that $\pi_{\sigma_j}$ weakly converges to the target

distribution $\pi$. In particular, we say that $\{Q_\sigma^g\}_\sigma$ is a locally or globally balanced proposal if the sequence $\sigma_j$ satisfies $\lim_j \sigma_j = 0$ or $\lim_j \sigma_j = \infty$, respectively.

**Locally Balanced Sampler**. Zanella (2020) showed that the locally balanced function $g(t) = tg(\frac{1}{t})$ defines the family of locally balanced proposals, which furthermore is asymptotically optimal for locally informed proposals with $\sigma \downarrow 0$. The most commonly used locally balanced function is $g(t) = \sqrt{t}$. Grathwohl et al. (2021) introduced a gradient approximation of the probability ratio $\pi(y)/\pi(x) \approx \exp((y-x)^\top \nabla \log \pi(x))$ to make the locally balanced proposal scalable. Sun et al. (2021); Zhang et al. (2022); Sun et al. (2022a); Rhodes & Gutmann (2022) show locally balanced sampler can be more efficient with large scale $\sigma$ and (Sun et al., 2022b) proves that the optimal scaling $\sigma$ for locally balanced proposal is achieved when the average acceptance rate is $0.574$.

## 3  ANY-SCALE BALANCED SAMPLER

Many recent work (Sun et al., 2021; Zhang et al., 2022; Sun et al., 2022b) have shown using larger kernel $K_\sigma$ can significantly improve the sampling efficiency in locally balanced samplers. Unfortunately, in migrating locally balanced samplers to a global proposal regime, these works ignore the fact that a locally balanced weight function is no longer optimal and the accuracy of the gradient approximation diminishes. To address such shortcomings, we propose our any-scale balanced samplers. We will consider sampling from the discrete space $\mathcal{X} = \mathcal{S}^d = \{1, ..., S\}^d$ with a target distribution $\pi(x) \propto e^{f(x)}$.

### 3.1  ANY-SCALE BALANCED FUNCTIONS

The first challenge in developing a non-local proposal is that the locally balanced weight function is no longer optimal. To determine the proper choice of weight function for kernels $K_\sigma = \exp(-\|z\|^2/2\sigma)$ at different scales, we examine the acceptance rate for the informed proposal in (1) and consider the simple but sufficiently representative weight function class $g(t) = t^\alpha$ for different $\alpha$. Note that, given a current state $x$ and new state $y$, we have the ratio

$$A_\sigma = \frac{\pi(y)Q_\sigma(y,x)}{\pi(x)Q_\sigma(x,y)} = \frac{\pi(y)(\frac{\pi(x)}{\pi(y)})^\alpha K_\sigma(x-y)/\sum_z (\frac{\pi(z)}{\pi(y)})^\alpha K_\sigma(z-y)}{\pi(x)(\frac{\pi(y)}{\pi(x)})^\alpha K_\sigma(y-x)/\sum_z (\frac{\pi(z)}{\pi(x)})^\alpha K_\sigma(z-x)} \tag{2}$$

$$= \frac{\pi^{1-\alpha}(y)/\sum_z \pi^\alpha(z)K_\sigma(z-y)}{\pi^{1-\alpha}(x)/\sum_z \pi^\alpha(z)K_\sigma(z-x)} = \frac{\pi^{1-\alpha}(y)/(\pi^\alpha * K_\sigma)(x)}{\pi^{1-\alpha}(x)/(\pi^\alpha * K_\sigma)(y)}, \tag{3}$$

where $(F * G)(x) = \sum_z F(z)G(x-z)$ represent the convolution of two functions. Based on this formulation, one can easily recover the locally balanced function. In particular, consider the local proposal at diminishing scales $\sigma$, leading to:

$$\lim_{\sigma \to 0}(\pi^\alpha * K_\sigma)(x) = \pi^\alpha(x) \quad \Rightarrow \quad \lim_{\sigma \to 0} A_\sigma = \pi^{1-2\alpha}(y)/\pi^{1-2\alpha}(x). \tag{4}$$

The limit ratio implies that $\alpha = \frac{1}{2}$ makes the stationary distribution of $\pi_\sigma$ weakly converge to the target distribution $\pi$; hence the corresponding weight function is $g(t) = \sqrt{t}$, which is one of the most widely used locally balanced functions (Zanella, 2020).

A more interesting question is how to select $\alpha$ for $\sigma > 0$. Since computing the convolution for a general target distributions is intractable, we consider a continuous relaxation to obtain a hint for determining the proper value of $\alpha$ for a given $\sigma > 0$. In particular, consider a normal target distribution $\pi(\cdot) \sim \mathcal{N}(\mu, \sigma_0 I)$ in the real space $\mathbb{R}^d$. In this case, the ratio has a closed form:

$$\pi^\alpha * K_\sigma \sim \mathcal{N}(\mu, (\sigma + \sigma_0/\alpha)I) = \pi^{\frac{\alpha\sigma_0}{\alpha\sigma+\sigma_0}} \quad \Rightarrow \quad A_\sigma = \pi^{1-\alpha-\frac{\alpha\sigma_0}{\alpha\sigma+\sigma_0}}(y)/\pi^{1-\alpha-\frac{\alpha\sigma_0}{\alpha\sigma+\sigma_0}}(x). \tag{5}$$

Here, to make the proposal balanced, the parameter $\alpha$ needs to satisfy

$$1 - \alpha - \frac{\alpha\sigma_0}{\alpha\sigma + \sigma_0} = 0 \quad \Rightarrow \quad \alpha = \frac{r - 2 + \sqrt{r^2 + 4}}{2r}, \quad r = \frac{\sigma}{\sigma_0}. \tag{6}$$

One can easily check that this family of balanced functions forms a set of interpolants between $g(t) = \sqrt{t}$ and $g(t) = t$, where the two limiting values are:

$$\lim_{\sigma \to 0} \alpha = \lim_{r \to 0} \frac{r - 2 + \sqrt{r^2 + 4}}{2r} = \frac{1}{2}, \quad \lim_{\sigma \to \infty} \alpha = \lim_{r \to \infty} \frac{r - 2 + \sqrt{r^2 + 4}}{2r} = 1. \tag{7}$$

The first equation recovers the locally balanced function $g(t) = \sqrt{t}$. The second equation shows that, if we consider all states as candidates in the proposal distribution, the optimal choice of weight function is $g(t) = t$, which causes the proposal distribution to become:

$$\lim_{\sigma \to \infty} Q_\sigma(x, y) = \frac{\pi(y)/\pi(x)}{\sum_{z \in \mathcal{X}} \pi(z)/\pi(x)} = \pi(y). \tag{8}$$

That is, the proposal degenerates to the target distribution. Ignoring computational cost, such a Markov chain draws independent samples from the target distribution in each step and has, in general, the best efficiency one can expect. Between these two limiting cases, the parameter $\alpha \in (0, 1)$ specifies an interpolation that needs to be carefully selected based on $\sigma$ to balance the proposal.

To this end, we employ an adaptive algorithm to automatically learn the proper configuration during sampling (Andrieu & Thoms, 2008). Since the hyperparameter pair $(\sigma, \alpha)$ is highly correlated, it can be challenging to directly tune them, so we instead employ a coordinate descent style method that alternatively updates $\sigma$ and $\alpha$ based on the average jump distance. Specifically, we probe the value with $(1 + \gamma)\sigma, \sigma, (1 - \gamma)\sigma$ with fixed $\alpha$ and select the new value of $\sigma$ based on which one has the largest average jump distance. And similar method to $\alpha$; see the Appendix A.2 for full details of the adaptation algorithm in Algorithm 3. In our experiments below, we observe that the effective sample size is a concave function of $\alpha$ for fixed $\sigma$ and vice versa, hence the adaption algorithm is typically able to find a good $(\sigma, \alpha)$ configuration efficiently.

## 3.2 QUADRATIC APPROXIMATION

The second challenge is that, in a non-local proposal, the gradient approximation of the probability ratio becomes less accurate. To capture the correlation between variables, we consider a quadratic approximation of the log probability change:

$$f(y) - f(x) = (y - x)^\top \nabla f(x) + \frac{1}{2}(y - x)^\top W(y - x). \tag{9}$$

When $W = \nabla^2 f(x)$ is the Hessian matrix, (9) becomes a second order Taylor approximation. In this work, we employ a global $W$ for all states $x$, hence we choose $W$ as the empirical average Hessian. In particular, the pairs $(y - x, \nabla g(y) - \nabla g(x))$ are first collected during a burn-in period and $W$ is selected as:

$$W = \arg\min_{W = W^\top} \sum_{i=1}^N \|W(y_i - x_i) - (\nabla f(y_i) - \nabla f(x_i))\|^2, \tag{10}$$

which can be efficiently solved via gradient descent. Please refer to Appendix A.3 for details. Substituting the quadratic approximation (9), into the informed proposal in (1), with weight function $g(t) = t^\alpha$, the quadratic proposal distribution becomes

$$Q(x, y) \propto \exp\left(\alpha[(y - x)^\top \nabla f(x) + \frac{1}{2}(y - x)^\top W(y - x)] - \frac{1}{2\sigma}(y - x)^\top(y - x)\right). \tag{11}$$

Although the second order approximation improves proposal quality from the perspective of gradient approximation, it also makes (11) become a pairwise Markov random field, which is typically intractable to sample from (Murray, 2007). Therefore, to develop a practical sampling algorithm, we exploit a stochastic factorization of quadratic proposal distribution known as the Gaussian integral trick, which originated in statistical physics (Hubbard, 1959; Hertz et al., 1991) and has been more recently extended in machine learning (Martens & Sutskever, 2010; Zhang et al., 2012). The original Gaussian integral trick is designed for binary random variables, here we show it also works on more general discrete random variables. In particular, for a quadratic distribution $\pi(z) \propto \exp(\frac{1}{2}z^\top Wz + z^\top b)$, and a PSD diagonal matrix $D$ that guarantees $W + D$ is PSD, one can introduce a Gaussian auxiliary variable $Q(u|z) \sim \mathcal{N}((W + D)^{\frac{1}{2}}z, I)$ so that the conditional distribution of $z$ given $u$ can be obtained via Bayes' rule:

$$Q(z|u) \propto \exp\left(\frac{1}{2}z^\top Wz + z^\top b - \frac{1}{2}(u - (W + D)^{\frac{1}{2}}z)^\top(u - (W + D)^{\frac{1}{2}}z)\right) \tag{12}$$

$$\propto \exp\left(z^\top[(W + D)^{\frac{1}{2}}u + b] - \frac{1}{2}z^\top Dz\right), \tag{13}$$

where the square root of a matrix can be obtained by either Cholesky or eigen decomposition. More details of the Gaussian integral trick are provided in Appendix A.5; also see Zhang et al. (2012) for a good introduction. To use this trick in sampling from (11), we first sample the auxiliary variable $u$

based on the current state $Q(u|x) \sim \mathcal{N}((W+D)^{\frac{1}{2}}x, I)$, then propose $y$ according to

$$Q(y|x,u) \propto \exp\left(\alpha y^\top [\nabla f(x) - Wx + (W+D)^{\frac{1}{2}}u] - \frac{1}{2}y^\top(\alpha D + \frac{1}{\sigma})y\right), \qquad (14)$$

Note that the marginal distribution $\int Q(u|x)Q(y|x,u)du$ is exactly $Q(x,y)$ in (11), hence we call this a stochastic factorization. By introducing the auxiliary variable $u$, we avoid calculating the intractable partition function for (11). The M-H acceptance test for this auxiliary sampler is:

$$A(x,u,y) = \min\left\{1, \frac{\pi(y)Q(u|y)Q(x|y,u)}{\pi(x)Q(u|x)Q(y|x,u)}\right\}. \qquad (15)$$

Given $W$, a good choice for $D$ should give high acceptance rate and large-variance proposal distribution. However, directly maximizing the acceptance rate and proposal variance at the current sample with respect to $D$ is intractable, therefore, we construct a surrogate.

Consider a continuous relaxation $\pi(x) \propto \exp(\frac{1}{2}x^\top Wx)$, where one can use $\sigma = \infty$, $\alpha = 1$ and the Gaussian integral trick guarantees the acceptance rate is always 1. In this case, the sampling efficiency is only determined by the variance of the proposal distribution. For a current state $x$ and auxiliary variables $\xi$ and $\zeta \sim \mathcal{N}(0, I_d)$, denote $u = (W+D)^{\frac{1}{2}}x + \xi$, and observe

$$y - x = D^{-1}Wx + D^{-1}(W+D)^{\frac{1}{2}}\xi + D^{-\frac{1}{2}}\zeta, \qquad (16)$$

for new state $y$ in (14). One can compute the variance of the change $(y-x)$ in proposal distribution in closed-form:

$$\mathbb{E}_x\mathbb{E}_{\xi,\zeta}\left[\left((y-x) - \mathbb{E}[y-x]\right)\left((y-x) - \mathbb{E}[y-x]\right)^\top\right] = 2D^{-1}, \qquad (17)$$

which is totally determined by the diagonal matrix $D$. See Appendix A.4 for detailed derivation. Therefore, we would like to minimize the diagonal of $D$ for larger variance, thus better sample efficiency, while still keep $W + D$ a PSD matrix.

A common approach to determine diagonal matrix $D$ is $\lambda$-shift, where $D = \lambda I$ is used with $\lambda = \max\{\epsilon, -\lambda_{\min}(W)\}$, such that $\epsilon \geq 0$ is a threshold and $\lambda_{\min}(W)$ is the smallest eigenvalue of $W$ (Martens & Sutskever, 2010). However, such an isotropic choice can suppress movement in dimensions with large variance. For example, consider a special case where $W$ is a diagonal matrix with $W_{11} = -100$ and $W_{jj} = -1$ for $j = 2, ..., d$. Using $D = 100I$ restricts the variance in all dimensions to 0.01, which is inefficient.

Instead, since the quadratic term $W$ is known, one can improve sampling efficiency by a more careful choice of diagonal matrix $D$. For example, Zhang et al. (2012) claims that the convexity of the proposal distribution depends on the spectrum of $W + D$. Following this idea, a straightforward choice is to minimize the largest eigenvalue of $W + D$. However, instead of only considering one direction, empirically, we find it is better to maximize the harmonic mean of $D^{-1}$ in (17). Intuitively, the harmonic mean provides a balanced approach to maximizing the variance of the proposal distribution, as it maximizes variance in all dimensions and puts more weight in directions with smaller variance. Conveniently, recall

$$\arg\max_D \frac{1}{\sum_{i=1}^d 1/D_{ii}^{-1}} = \arg\max_D \frac{1}{\sum_{i=1}^d D_{ii}} = \arg\min_D \sum_{i=1}^d D_{ii} = \arg\min_D \mathrm{tr}(D), \qquad (18)$$

maximizing the harmonic mean of the variance can be reduced to minimizing the trace of $D$. Under the constraint that $W + D$ is PSD, we obtains a semi-definite programming (SDP) problem,

$$\min_D \mathrm{trace}(D) \quad \text{s.t. } D \succeq 0, \ W + D \succeq 0 \qquad (19)$$

Empirically, we found there is no need for an exact optimum for (19); a rough solution after early stopping is sufficient to characterize the variance scale in each dimension. Using a modern solver this estimation step can be typically done in milliseconds for domain with 100 dimensions.

**Complete Algorithm**. With the any-scale wight function and the quadratic approximation, we are ready to present our *any-scale balanced sampler* (AB sampler) in Algorithm 1. The parameters $(\sigma, \alpha)$ are automatically tuned along the way of sampling (also see Algorithm 3 and Algorithm 4 for the details on adaptive tuning of these parameters), where $W$ and $D$ are 0 during burn-in, and updated via (10) and (18) right after burn-in.

| **Algorithm 1:** AB sampling algorithm | **Algorithm 2:** AB M-H step |
|---|---|
| **Input:** Initial $\sigma = 0.1, \alpha = 0.5, W = 0, D = 0$; initial $x_0$ 
 **Output:** MCMC chain $x_{0:T}$ and adjusted $\sigma, \alpha, W, D$ 
 1 **Burn-in period** $t < T_1$: alternatively update $\sigma, \alpha$ use algorithm 3 while calling M-H step defined in algorithm 2 as subroutine; collect trace $x_{0:T_1}$ along the way 
 2 **Estimate** $W$ and $D$ using collected $x_{0:T_1}$ via (10) and (18) 
 3 **Mixed period** $T_1 \leq t \leq T$: use estimated $W$ and $D$ to continue the alternatively updating $\sigma, \alpha$ with algorithm 3, while calling M-H step defined in algorithm 2 as subroutine; 
 4 **Return** the entire trace $x_{0:T}$ and the estimated parameters. | **Input:** $\sigma, \alpha, W, D$; current state $x_t$ 
 **Output:** new state $x_{t+1}$ 
 1 Sample auxiliary: 
 $u \sim \mathcal{N}((W+D)^{\frac{1}{2}}x_t, I)$ 
 2 Sample new state: $y \sim$ (14) 
 3 Get acceptance rate $A$ in equation (15) 
 4 **if** $rand(0,1) < A$ **then** 
 $x_{t+1} = y$ **else** $x_{t+1} = x_t$; |

## 4 RELATED WORK

The informed proposal, which uses information about the target distribution to guide the proposal for the Metropolis-Hastings (M-H) algorithm has been extensively studied for discrete spaces in recent years. A number of methods have attempted to first map the discrete to a continuous space, using relaxation, apply gradient based methods in the continuous space, then map the new state back to the discrete space, either by using auxiliary variables, uniform dequantization, or VAE flow (Zhang et al., 2012; Pakman & Paninski, 2013; Nishimura et al., 2017; Han & Liu, 2018; Zhou, 2020; Jaini et al., 2021). Such methods work in some scenarios, but embedding a discrete into a continuous space often destroys its natural topological structure, and can create highly multimodal and irregular target distributions (Zanella, 2020). As shown in previous work, such methods does not scale well to high dimensional discrete settings (Grathwohl et al., 2021).

Another group of methods attempt to directly work within discrete spaces. Titsias & Yau (2017) and Dai et al. (2020) introduce auxiliary variables to trade off the number of updated variables in a block against computational cost, however, by relying on Gibbs sampling, such methods still require significant overhead to make updates. In addition to the related works (Zanella, 2020; Grathwohl et al., 2021; Sun et al., 2021; 2022a; Zhang et al., 2022) already discussed in depth above, a concurrent work (Rhodes & Gutmann, 2022) has considered preconditioning and also used the Gaussian integral trick to incorporate second order information from the target distribution, but this work does not study how to properly choose the weight function $g$, the hyperparameter $(\sigma, \alpha)$, and the diagonal matrix $D$, making the resulting algorithm less efficient. Another recent work (Sun et al., 2022b) proves that the optimal scale $\sigma$ for locally balanced proposal is achieved when the average acceptance equals to $0.574$, and give a robust adaptive algorithm for tuning $\sigma$. However, its result relies on the property of locally balanced function, and does not apply to more general weight function $g(t) = t^\alpha$ with $\alpha \neq 0.5$.

## 5 EXPERIMENTS

We conducted an experimental evaluation on three types of target distributions: 1) quadratic synthetic distributions, 2) non-quadratic synthetic distributions, and 3) real distributions. For quadratic synthetic distributions, we focus on demonstrating the benefits of selecting a high quality diagonal matrix $D$. For non-quadratic synthetic distributions, we show that the performance of the proposed sampler significantly relies on the choice of weight function $g(t) = t^\alpha$. For real distributions, we compare against baseline samplers on challenging inference problems in deep energy based models trained on MNIST, Omniglot, and Caltech datasets.

### 5.1 SETTINGS

**Samplers**. We denote the proposed *Any-scale Balanced sampler* as AB-trace sampler, which uses the any-scale balanced function $g(t) = t^\alpha$ and obtains the diagonal matrix $D$ by minimizing its trace. For comparison, we consider the classical discrete samplers, random walk Metropolis (RWM) and Gibbs sampler. We also compare to a locally balanced sampler (LB), considering a representative version DLP in Zhang et al. (2022) that uses $\alpha = 0.5$ and $W = 0$ in (11); this is mathematically equivalent to NCG in Rhodes & Gutmann (2022). For RWM and LB, we follow the optimal acceptance rate in (Sun et al., 2022b) and tune the scale of the proposal distribution until the average acceptance is $0.234$ and $0.574$, respectively.

Table 1: ESS on selected Quadratic Distributions

| Distribution | Grid Ising | | BA-4 Ising | | Rotation Gaussian | | Sparse Gaussian | |
|---|---|---|---|---|---|---|---|---|
| Sampler | $\text{ESS}_n$ | $\text{ESS}_t$ | $\text{ESS}_n$ | $\text{ESS}_t$ | $\text{ESS}_n$ | $\text{ESS}_t$ | $\text{ESS}_n$ | $\text{ESS}_t$ |
| Gibbs | 1.66 | 0.55 | 8.53 | 2.84 | 2.81 | 11.26 | 4.20 | 16.81 |
| RW | 1.27 | 0.31 | 4.28 | 1.07 | 9.41 | 2.69 | 8.67 | 2.48 |
| DLP | 2.96 | 0.49 | 120.15 | 20.89 | 86.28 | 15.69 | 71.23 | 13.57 |
| DLP-trace | 2.88 | 0.47 | 143.25 | 24.90 | 86.62 | 15.75 | 142.86 | 27.21 |
| AB-1st | 3.99 | 0.67 | 242.31 | 35.89 | 103.16 | 16.50 | 79.88 | 13.31 |
| AB-shift (PAVG) | 7.87 | 1.12 | 274.18 | 39.16 | 196.88 | 30.29 | 154.00 | 24.64 |
| AB-max | 7.83 | 1.11 | 287.24 | 41.03 | 217.41 | 33.45 | 474.75 | 75.96 |
| AB-trace | **7.94** | **1.13** | **776.31** | **110.90** | **239.09** | **36.78** | **1227.94** | **196.47** |

To demonstrate the benefit of the proposed methods, we consider a few variants for ablation: DLP-trace, which uses the same anisotropic diagonal matrix $D$ as AB-trace in kernel $K_\sigma(z) = \exp(-z^\top D z/\sigma)$ for DLP, AB-1st, which only uses gradients to approximate the probability ratio, and AB-shift and AB-max, which obtain the diagonal matrix $D$ via $\lambda$-shift or minimizing the maximum eigenvalue of $W + D$, respectively. For all AB-* samplers, we tune $(\sigma, \alpha)$ adaptively via the algorithm discussed above (and described in Appendix A.2). Note that the PAVG sampler in (Rhodes & Gutmann, 2022) is equivalent to AB-shift with fixed $(\sigma, \alpha) = (\infty, 1)$. Since we find that tuning $(\sigma, \alpha)$ improves the efficiency, we use AB-shift to represent PAVG. More details about the sampler implementations, such as solving $D$, are given in Appendix A.1.

**Metrics**. As in other works (Hoffman et al., 2014; Zanella, 2020), we use effective sample size (ESS) (Lenth, 2001) to characterize the efficiency of the samplers on synthetic distributions. To reduce the effects of implementation, we report ESS normalized in two different ways: We let $\text{ESS}_n$ denote the ESS for every 10,000 queries of the log likelihood function, and $\text{ESS}_t$ denote the ESS for every one second of sampling. For each setting and sampler, we run 100 chains for $T$=100,000 steps, with $T_1$=20,000 burn-in steps to make sure the chain mixes. For real distributions, we compare the mixing time for different samplers.

## 5.2 QUADRATIC SYNTHETIC DISTRIBUTIONS

**Ising model**. The Ising model (Ising, 1924) is a mathematical model of ferromagnetism in statistical mechanics. It consists of binary random variables arranged in a graph $G = (V, E)$ and allows each node to interact with its neighbors. The unnormalized log probability function of the Ising model is:

$$f(x) = \sum_{i \in V} w_i x_i + \sum_{(i,j) \in E} J_{ij} x_i x_j. \tag{20}$$

In this experiment, we consider Ising models on 2D grid graphs and Barabasi-Albert-4 graphs (Albert & Barabási, 2002). For grid Ising, we set $J_{ij} = 0.4407$ at the critical temperature (Onsager, 1944), so that the model is at its transition phase and hard to sample from; see Appendix B.2 for a more detailed description. We conduct sampling at high, medium, and low temperatures. In Table 1, we report results for the medium temperature, where $J_{ij} = 0.4407$. More results on Ising model are given in Table 2 and Table 3.

**Lattice Gaussian Model**. The lattice Gaussian model is obtained by restricting the Gaussian distribution to a Lattice, which is an important distribution in coding and cryptography (Kschischang & Pasupathy, 1993; Micciancio & Regev, 2007). The unnormalized log probability is:

$$f(x) = -\tfrac{1}{2}(x - b)^\top W (x - b). \tag{21}$$

In this experiment, we use a finite state space $\mathcal{X} = \{0, 1, ..., 20\}^{100}$ and we investigate two settings for the Gaussian model. The first setting is a rotated Gaussian $W = P^\top \Lambda P$, where $P$ is an orthogonal matrix and $\Lambda$ is a diagonal matrix. The second setting is a Sparse Gaussian, which is a pairwise Markov random field defined on a cycle. We constructed the lattice Gaussian models with low, medium, and high conditions. More detailed descriptions of these models are given in Appendix B.3. We report the results for medium condition in Table 1. More results are given in Table 4 and 5.

**Results Analysis**. One can observe that the AB samplers substantially outperform existing samplers on all distributions. Specifically, the first order sampler AB-1st has ESS consistently larger than LB, which justifies the benefit of selecting the proper weight function. Also, the AB-trace sampler has comparable efficiency to AB-shift on Grid Ising and Rotation Gaussian, but is significantly better on

| Sampler | ESS$_n$ | ESS$_t$ |
|---|---|---|
| Gibbs | 43.81 | 14.60 |
| RW | 25.00 | 6.25 |
| DLP | 222.49 | 35.60 |
| DLP-trace | 247.35 | 40.69 |
| AB-1st | 321.56 | 51.44 |
| AB-shift (PAVG) | 265.89 | 37.98 |
| AB-max | 266.49 | 38.06 |
| AB-trace | **403.78** | **57.68** |

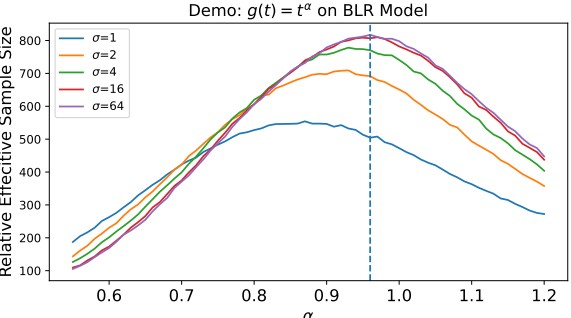

Figure 2: Results on BLR: (l) ESS for different samplers, (r) ESS for AB-trace with different $(\sigma, \alpha)$

| Sampler | ESS$_n$ | ESS$_t$ |
|---|---|---|
| Gibbs | 13.66 | 34.14 |
| RW | 189.69 | 34.49 |
| DLP | 97.97 | 12.25 |
| DLP-trace | 100.65 | 12.58 |
| AB-1st | 256.88 | 32.11 |
| AB-shift (PAVG) | 327.81 | 36.42 |
| AB-max | 355.78 | 39.53 |
| AB-trace | **380.50** | **42.28** |

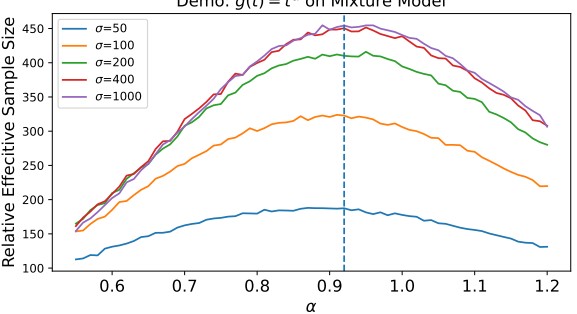

Figure 3: Results on QMM: (l) ESS for different samplers, (r) ESS for AB-trace with different $(\sigma, \alpha)$

BA-4 Ising and Sparse Ising. The reason is that the variables in Grid Ising and Rotation Gaussian are nearly homogeneous, and an isotropic diagonal matrix $D = \lambda I$ is not slowed by several hard dimensions. However, in BA-4 Ising and Sparse Gaussian, the variance in different dimensions can be very different, and AB-trace demonstrates significant advantages by employing a general diagonal matrix $D$ that allows different step sizes in different dimensions.

## 5.3 NON-QUADRATIC SYNTHETIC DISTRIBUTIONS

**Bayesian Logistic Regression (BLR)**. Following Zhou (2020), we consider a logistic regression model $Y \sim \text{Bernoulli}(\text{sigmoid}(X\beta))$, with $Y \in \{0,1\}^m, X \in \mathbb{R}^{m \times d}, \beta \in \{0,1\}^d$. We first generate the sample $X, Y$, then, using a uniform prior, the target distribution is the posterior of $\beta$ with the unnormalized log probability function:

$$f(\beta) = -\sum_{i=1}^m y_i \log\left(1 + \exp(-\sigma_i)\right) + (1 - y_i) \log\left(1 + \exp(\sigma_i)\right), \quad \sigma_i = \sum_{j=1}^d X_{ij}\beta_j. \quad (22)$$

In this experiment, we considered a $d = 100$ dimensional regression with $m = 50$ samples. More details for generating $X, Y$ are given in Appendix B.4. We report the results for ESS in Table 2. For AB-trace, the selected configuration is $(\sigma, \alpha) = (16, 0.96)$. To justify the quality of this selection, we also plot the ESS for AB-trace with different $(\sigma, \alpha)$ in Figure 2.

**Quartic Mixture Model (QMM)**. Following Rhodes & Gutmann (2022), we consider a quartic mixture model, where the unnormalized log likelihood function can be written as:

$$f(x) = \log\left(\sum_{k=1}^K \exp(-\text{poly}_k^4(x))\right). \quad (23)$$

such that $\text{poly}_k^4$ is multivariate polynomial with degree 4. In this experiment, we use a finite state space $\mathcal{X} = \{0, 1, ..., 20\}^{50}$ and $K = 50$ components for the mixture model. More details about $\text{poly}_k^4$ are given in Appendix B.5. We report the ESS results in Figure 3. For AB-trace, the selected configuration is $(\sigma, \alpha) = (415, 0.92)$. To justify the quality of this selection, we also plot the ESS for AB-trace with different $(\sigma, \alpha)$ in Figure 3

**Results Analysis**. For the non-quadratic synthetic distributions, the AB-trace sampler significantly outperformed the other methods. From the curves for $(\sigma, \alpha)$, one can see that the adaptive tuning algorithm successfully found optimal configurations. Note that in Figure 2 and Figure 3, the values $\sigma = 64$ and $\sigma = 1000$ can be seen as infinity, since further increasing the $\sigma$ does not influence

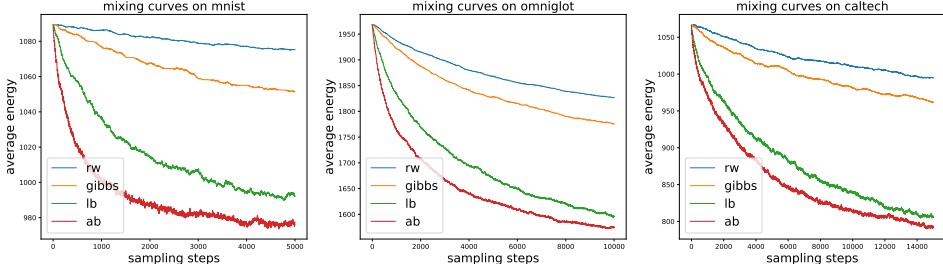

Figure 4: Mixing Time on Real Distributions

efficiency. Unlike from (8), the optimal $\alpha$ are still not 1 as we have some estimation error for the probability ratio. One interesting phenomenon is that the first order method AB-1st can be more efficient than second order samplers AB-shift and AB-max in BLR. The reason is that the Gaussian integral trick introduces extra variance in the proposal distribution. If the diagonal matrix $D$ is not properly selected, the benefit of using a second order sampler can be reduced.

### 5.4 DEEP EBMs ON REAL DISTRIBUTIONS

Having observed excellent performance of the AB sampler on synthetic datasets, we considered sampling in more challenging real distributions. In particular, here we trained deep EBMs parameterized by ResNet (He et al., 2016) on the MNIST, Omniglot, and Caltech datasets. In these real image distributions, we are interested in how fast sampling algorithms can find high quality images, so we report the mixing rate in figure 4. Since we are comparing behavior during the mixing stage, we do not have samples to estimate $W$ via (10), hence we use AB sampler with a bit different from Algorithm 1. In particular, we use the true data (from datasets) to estimate the variance $\text{var}_i$ for each variable $x_i$ and set $W$ as a diagonal matrix with $W_{ii} = 1/(1 + \text{var}_i)$. In this case, we do not need to use the Gaussian integral trick and we do not distinguish the different version of AB samplers. More details are given in Appendix B.6.

In Figure 4, one can see that the AB sampler mixes faster than the LB sampler on all three real distributions. The optimal $\alpha$ selected for MNIST, Omniglot, and Caltech are $0.6, 0.55, 0.55$, respectively. They are significantly smaller than that in non-quadratic synthetic distributions. We believe the reason is that these deep EBMs are much more complicated than the synthetic distributions. Larger estimation errors only allow the sampler to make local movements, and hence have a smaller $\alpha$.

## 6 CONCLUSION

In this work, we proposed an *Any-scale Balanced sampler (AB sampler)* that substantially improves existing locally balanced samplers for discrete spaces in two respects:

- the AB sampler goes beyond considering the locally balanced function as an "optimal" choice for weight function in an informed proposal, and provides an adaptive algorithm for finding the optimal configuration of $(\sigma, \alpha)$;
- the AB sampler introduces the Gaussian integral trick, which allows efficient second order approximation to improve proposal quality.

There are still directions for further improvement of the AB sampler. First, current adapting Algorithm 3 tunes $(\sigma, \alpha)$ based on the empirical estimation of jump distance, which can vary a lot during the mixing process. As a result, our adapting algorithm is not stable until the Markov chain reaches its stationary distribution. This is not a big problem in sampling. But if we want to train EBMs via contrastive divergence (Hinton, 2002; Tieleman, 2008), the current adapting algorithm can hardly find the optimal configuration for $(\sigma, \alpha)$ as the model keeps changing. A potential solution is to use adaptive algorithms based on acceptance rate (Roberts & Rosenthal, 2001; Sun et al., 2022b). Second, the estimation of $W$ in (10) is rough. In complicated distributions, the Hessian matrix can vary a lot at different state $x$. More accurate quadratic approximation can be obtained via Riemannian (Girolami & Calderhead, 2011) or quasi-Newton style algorithms (Zhang & Sutton, 2011) that allow $W = W(x)$ depend on the current state $x$. Third, the current quadratic approximation can be inefficient on large models. For example, on a large sparse quadratic model, solving the sparse SDP can be time consuming, and the square root $(W + D)^{\frac{1}{2}}$ is not necessarily sparse. Hence, more sophisticated design are needed to make the quadratic approximation being scalable.

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

# A SAMPLERS

## A.1 SEMI-DEFINITE PROGRAMMING

Solving the diagonal matrix $D$ for AB-trace and AB-max can be formulated as semi-definite programming (SDP). In particular, for AB-trace, the SDP problem is:

$$D^* = \arg\min_D \; \text{trace}(D) \tag{24}$$

$$\text{s.t. } D \succeq 0, \; W + D \succeq 0 \tag{25}$$

For AB-max, the SDP problem is

$$D^* = \arg\min_D \; \lambda \tag{26}$$

$$\text{s.t. } D \succeq 0, \; \lambda I \succeq W + D \succeq 0 \tag{27}$$

Both SDP problems can be efficiently solved by modern SDP solver. In this work, we use academia version of Mosek (ApS, 2019). For models have less or equal to 100 variables (Lattice Gaussian, Bayesian Logistic Regression, Quartic Mixture Model), Mosek takes less than 0.05 second to solve the SDP problem. For models with 400 variables (ISing), Mosek took less than 1 second to solve the SDP problem. For models with 784 variables, Mosek takes around 10 seconds to solve the SDP problem. For all distributions considered in this work, the time used for solving SDP is negligible comparing to the sampling time. However, for models with several thousands or more variables, the cost for directly solving the SDP could be high and better methods to estimate the diagonal matrix $D$ are needed.

## A.2 ADAPTIVE TUNING ALGORITHM

We give the pseudo code for adaptive tuning of $(\sigma, \alpha)$ in Algorithm 3. The basic idea is alternatively updating $\sigma$ and $\alpha$ to maximizing the average jump distance. In line 1, 2, and 3 in Algorithm 4, the samples are collected via calling M-H step of AB sampler as in Algorithm 2.

---

**Algorithm 3:** Adapting Algorithm

**Input:** initial $\sigma = 0.1, \alpha = 0.5$, update rate $\gamma = 0.2$, decay rate $\beta = 0.9$, initial state $x_0$, buffer size $N = 100$.
**Output:** parameters $\sigma, \alpha$, samples $x_1, x_2, ...$

1 **for** $i = 0, 1, ...$ **do**
2 $\quad \sigma', (x_{6iN+1}, ..., x_{6iN+3N}) \leftarrow$ Adapting Algorithm Block$(\theta = \sigma, \gamma, x_0)$
3 $\quad \alpha', (x_{6iN+3N1}, ..., x_{6iN+6N}) \leftarrow$ Adapting Algorithm Block$(\theta = \alpha, \gamma, x_0)$
4 **end**
5 **if** $\sigma == \sigma', \alpha == \alpha'$ **then**
6 $\quad \gamma = \beta\gamma$
7 **else**
8 $\quad \sigma = \sigma', \alpha = \alpha'$
9 **end**

---

## A.3 QUADRATIC APPROXIMATION

Here, we explain how to efficiently solve the following optimization problem:

$$W^* = \arg\min_{W=W^\top} \sum_{i=1}^N \|W(y_i - x_i) - (\nabla f(y_i) - \nabla f(x_i))\|^2. \tag{28}$$

Denote the $y_i - x_i$ forms a matrix $X\mathbb{R}^{N \times d}$, such that the $i$-th row $M_i = y_i - x_i$. Similarly, denote the $\nabla f(y_i) - \nabla f(x_i)$ forms a matrix $Y \in \mathbb{R}^{N \times d}$, such that the $i$-th row $Y_i = \nabla f(y_i) - \nabla f(x_i)$. Then, the loss function can be rewritten as:

$$W^* = \arg\min_{W=W^\top} \sum_{j=1}^d \|XW_{:,j} - Y_{:,j}\|_2^2 \tag{29}$$

---

**Algorithm 4:** Adapting Algorithm Block

**Input:** target parameter $\theta$, adapting rate $\gamma$, initial state $x_0$
**Output:** updated parameter $\theta'$, samples $x_1, ..., x_{3N}$

1   Using parameter $\theta$ to sample $x_1, ..., x_N$ via Algorithm 2
2   Compute $d_0 = \sum_{i=1}^{N} |x_i - x_{i-1}|_1$
3   Using parameter $\theta(1 + \gamma)$ to sample $x_{N+1}, ..., x_{2N}$ via Algorithm 2
4   Compute $d_+ = \sum_{i=1}^{N} |x_{N+i} - x_{N+i-1}|_1$
5   Using parameter $\theta(1 - \gamma)$ to sample $x_{2N+1}, ..., x_{3N}$ via Algorithm 2
6   Compute $d_- = \sum_{i=1}^{N} |x_{2N+i} - x_{2N+i-1}|_1$
7   **if** $\max\{d_0, d_+, d_-\} == d_+$ **then**
8     |   $\theta' = \theta(1 + \gamma)$
9   **else if** $\max\{d_0, d_+, d_-\} == d_-$ **then**
10   |   $\theta' = \theta(1 - \gamma)$
11   **else**
12   |   $\theta' = \theta$
13   **end**

---

where $W_{:,j}$ represents the $j$-th column of $W$. One can easily see that the loss function is a regression. For the feasible region $W = W^\top$, one can easily check it is a $\frac{d(d+1)}{2}$ dimensional linear subspace. As a result, one only efficiently solving this convex optimization problems via projected gradient descent, where the projection to symmetric matrix space is simply $X \to \frac{X+X^T}{2}$.

### A.4   VARIANCE OF THE PROPOSAL DISTRIBUTION

For $\xi$ and $\zeta \sim \mathcal{N}(0, I_d)$, $u = (W + D)^{\frac{1}{2}} x + \xi$, and

$$y - x = D^{-1}Wx + D^{-1}(W + D)^{\frac{1}{2}}\xi + D^{-\frac{1}{2}}\zeta,$$

we have

$$\mathbb{E}_x \mathbb{E}_{\xi, \zeta} \left[ \left( (y - x) - \mathbb{E}[y - x] \right) \left( (y - x) - \mathbb{E}[y - x] \right)^\top \right] \tag{30}$$

$$= \mathbb{E}_x [D^{-1}Wxx^T WD^{-1} + D^{-1}(W + D)D^{-1} + D^{-1}] \tag{31}$$

$$= -D^{-1}WD^{-1} + D^{-1}WD^{-1} + 2D^{-1} \tag{32}$$

$$= 2D^{-1} \tag{33}$$

where (32) is because for a normal random variable $x \propto \exp(\frac{1}{2}x^T Wx)$, the variance of $x$ is $-W^{-1}$.

### A.5   GAUSSIAN INTEGRAL TRICK

The Gaussian integral trick, also known as Hubbard-Stratonovich transform (Hubbard, 1959), is first named in (Hertz et al., 1991) and extended by Martens & Sutskever (2010); Zhang et al. (2012) for efficient Gibbs/HMC sampling inference. The main idea is for discrete-valued pairwise-MRF with within-layer connections, by introducing a real-valued auxiliary variable, the quadratic form in the energy function, $x^\top Wx$, will be canceled out. Thus, the inference will be easy to carry on. Specifically, we would like to sample from a MRF with pairwise dependency, *i.e.*,

$$p(x) = \exp(\frac{x^\top Wx}{2} + x^\top b)/Z, \tag{34}$$

where $x \in \{0, 1\}^d$ and $Z = \sum_x \exp(x^\top Wx + x^\top b)$. The vanilla sampler for this model is Markov chain Monte Carlo or Gibbs sampling, which although provably converges to the target distribution, but still might stuck in some region.

To accelerate the sampling, one can introduce the auxiliary variables to reformulate the MRF into a family of equivalent Boltzmann machines. Concretely, we introduce the auxiliary variable $u$ with conditional distribution:

$$p(u|x) = \mathcal{N}(u|A(W + D)x, A(W + D)A^\top), \tag{35}$$

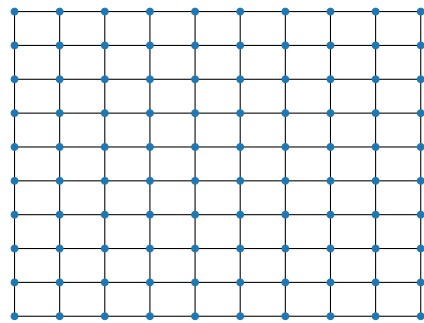 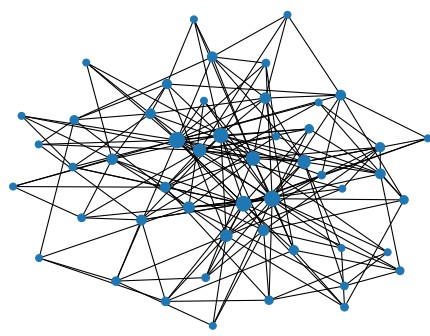

Figure 5: Visualization: (l) $10 \times 10$ Grid Graph, (r) 50-4 BA Graph

where $D = \mathrm{diag}(d)$. Therefore, we have the joint distribution as

$$p(x, u) \propto \exp\left(-\frac{u^\top \left(A^{-1}\right)^\top (W + D)^{-1} A^{-1} u}{2} + x^\top A^{-1} u + \left(b - \frac{1}{2}d\right)^\top x\right), \qquad (36)$$

which cancels the quadratic term w.r.t. $x$ and makes $x$ independent for each dimension. This induces the $p(x|u)$ is a multivariate Bernoulli distribution, *i.e.*,

$$p(x|u) = \prod_{i=1}^{d} (1 - p_i)^{1 - x_i} \, p_i^{x_i}, \qquad (37)$$

with $p_i = \frac{1}{1 + \exp(-t_i)}$, $t_i = b_i - \frac{1}{2}d_i + (A^{-1}u)_i$.

One shall notice that in the original Gaussian integral trick (36), the simplification $-\frac{1}{2}x^\top D x = -\frac{1}{2}d^\top x$ using the property that $x$ are binary random variables. Actually, we don't have to use this simplification. As long as $D$ is diagonal, we can factorize the distribution and make efficinet proposal. In this work, we consider using $A = (W + D)^{-\frac{1}{2}}$, such that the conditional distribution has following simple forms:

$$p(u|x) = \mathcal{N}(u|(W + D)^{\frac{1}{2}}x, I), \qquad (38)$$

$$p(x|u) \propto \exp\left(x^\top ((W + D)^{\frac{1}{2}}u + b) - \frac{1}{2}x^\top D x\right). \qquad (39)$$

## B  EXPERIMENTS

### B.1  HARDWARE

All experiments are running on a virtual machine with CPU: Intel Haswell, GPU: $4\times$ Nvidia V100, System: Debian 10.

### B.2  ISING MODEL

The Ising model (Ising, 1924) is a mathematical model of ferromagnetism in statistical mechanics . It consists of binary random variables arranged in a graph $G = (V, E)$ and allows each node to interact with its neighbors. The log probability function of Ising model is:

$$f(x) = \sum_{i \in V} w_i x_i + \sum_{(i,j) \in E} J_{ij} x_i x_j \qquad (40)$$

In this experiment, we consider Ising models on 2D grid graphs and Barabasi-Albert graphs (Albert & Barabási, 2002).

**grid Ising**. We consider $20 \times 20$ grid graphs. See Figure 5 for visualization of a $10 \times 10$ grid graph. We use zero external force $w_i = 0$ and set the interaction $J_{ij} = 0.3000, 0.4407, 0.7071$ for high,

critical, and low temperatures. We report the results in Table 2. Consistent to the results in statistical physics, phase transition occurs at the critical temperature and makes the sampling much harder (Onsager, 1944). More detailedly, at low temperature, variables in grid Ising model only have strong correlation with variables close to it. At critical temperature, the correlation is global and a variable can strongly depends on variables far from it. At high temperature, the variables have weak correlation to all the other variables. On all these three scenarios, Any-scale (AB) samplers substantially outperforms previous discrete sampling methods, including locally balanced (LB) samplers.

Table 2: ESS on $20 \times 20$ Grid Ising

| Distribution | Interaction (0.3000) | | Interaction (0.4407) | | Interaction (0.7071) | |
|---|---|---|---|---|---|---|
| Sampler | $\text{ESS}_n$ | $\text{ESS}_t$ | $\text{ESS}_n$ | $\text{ESS}_t$ | $\text{ESS}_n$ | $\text{ESS}_t$ |
| Gibbs | 4.50 | 1.50 | 1.66 | 0.55 | 6.11 | 2.04 |
| RW | 4.81 | 1.20 | 1.27 | 0.31 | 2.02 | 0.50 |
| LB | 43.16 | 6.91 | 2.96 | 0.49 | 62.06 | 10.34 |
| AB-1st | 69.91 | 11.19 | 3.99 | 0.67 | 78.44 | 13.07 |
| AB-shift | 178.70 | 24.65 | 7.87 | 1.12 | 111.94 | 15.44 |
| AB-max | 178.68 | 24.65 | 7.83 | 1.11 | 117.06 | 16.15 |
| AB-trace | **182.91** | **65.23** | **7.94** | **1.13** | **149.06** | **20.56** |

**BA Ising**. We consider 400-4 BA graph, that's to say, a Barabasi-Albert random graph with 400 nodes and 4 attach edges for every node (Albert & Barabási, 2002). See Figure B.2 for visualization of a 50-4 BA graph. Since we don't know the critical temperature in BA graphs, we keep using the settig in grid Ising with zero external force $w_i = 0$ and interaction $J_{ij} = 0.3000, 0.4407, 0.7071$ for high, critical, and low temperatures. We report the results in Table 3. On all three temperatures, Any-scale (AB) samplers substantially outperforms previous discrete sampling methods, including locally balanced (LB) samplers. Also, one can notice that AB-trace sampler significantly outperforms other AB samplers using quadratic approximation. In low temperature model Ising (0.7071), the first order method BA-1st even beat AB-shift and AB-max using quadratic approximation. The reason is that the variables in BA graph has inhomogeneous topology and a casual selection of $D$ does help. Furthermore, using quadratic approximation has to involve extra randomness in Gaussian integral trick and harm the proposal quality. In grid graphs where different variables have very similar topology, thus this drawback is less significant.

Table 3: ESS on 400-4 BA Ising

| Distribution | Interaction (0.3000) | | Interaction (0.4407) | | Interaction (0.7071) | |
|---|---|---|---|---|---|---|
| Sampler | $\text{ESS}_n$ | $\text{ESS}_t$ | $\text{ESS}_n$ | $\text{ESS}_t$ | $\text{ESS}_n$ | $\text{ESS}_t$ |
| Gibbs | 8.19 | 3.27 | 8.53 | 2.84 | 8.75 | 3.50 |
| RW | 7.07 | 2.02 | 4.28 | 1.07 | 3.06 | 0.88 |
| LB | 117.06 | 20.36 | 120.15 | 20.89 | 153.16 | 26.64 |
| AB-1st | 207.91 | 36.16 | 242.31 | 35.89 | 354.50 | 61.65 |
| AB-shift | 367.72 | 54.48 | 274.18 | 39.16 | 160.03 | 23.71 |
| AB-max | 402.06 | 59.56 | 287.24 | 41.03 | 172.47 | 25.55 |
| AB-trace | **779.12** | **115.43** | **776.31** | **110.90** | **848.72** | **124.85** |

## B.3 LATTICE GAUSSIAN

**Rotation Gaussian**. For rotation Gaussian, we use bias vector $b = 0$. Given parameter $L$, we generate the weight matrix $W$ in the following way: We first generate the diagonal matrix $\Lambda \in \mathbb{R}^{100 \times 100}$, such that

$$\Lambda_{ii} = \frac{\sqrt{L}}{1 + (L - 1) * (i - 1)/99}, \quad i = 1, 2, ..., 100 \tag{41}$$

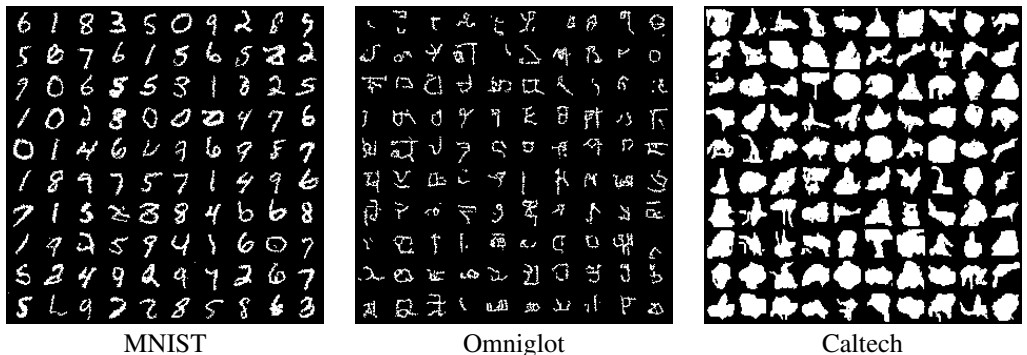

MNIST                   Omniglot                   Caltech

Figure 6: Images sampled from the trained EBMs.

Then we generate orthogonal matrix $P$ and let $W = -P\Lambda P^\top$. The results for rotation Gaussian with $L = 2, 10, 50$ are reported in Table 4.

**Sparse Gaussian** For Sparse Gaussian, we use bias vector $b = 0$. Given parameter $l$, we generate the weight matrix $W$ in the following way: We first generate the matrix $M \in \mathbb{R}^{100 \times 100}$, with $W_{ii} = 1$ and $W_{i,i+1} = W_{i+1,i} \sim \mathcal{N}(0, 0.04)$, for $i = 1, 2, ..., 100$. One shall notice that we denote $W_{100,101} = W_{100,1}$ and $W_{101,100} = W_{1,100}$. Then we generate the diagonal matrix $\Lambda$, such that

$$\Lambda_{ii} = \left(\frac{99}{1 + i * (L - 1)}\right)^{\frac{1}{2}}, \quad i = 1, 2, ..., 100 \tag{42}$$

Then, we let $W = -\Lambda M \Lambda$. Such a log probability function defines a graphical model defined on a cycle. Since a cycle is sparse, we call it sparse Gaussian. The results for sparse Gaussian with $L = 2, 10, 50$ are reported in Table 4.

Table 4: ESS on 100d Rotation Gaussian

| Distribution | $L = 2$ | | $L = 10$ | | $L = 50$ | |
|---|---|---|---|---|---|---|
| Sampler | $\mathrm{ESS}_n$ | $\mathrm{ESS}_t$ | $\mathrm{ESS}_n$ | $\mathrm{ESS}_t$ | $\mathrm{ESS}_n$ | $\mathrm{ESS}_t$ |
| Gibbs | 4.55 | 1.82 | 2.81 | 11.26 | 1.64 | 6.58 |
| RW | 14.53 | 4.15 | 9.41 | 2.69 | 6.20 | 1.77 |
| LB | 222.47 | 42.38 | 86.28 | 15.69 | 27.28 | 5.20 |
| AB-1st | 492.12 | 93.74 | 103.16 | 30.29 | 27.97 | 5.33 |
| AB-shift | 1182.16 | 189.15 | 196.88 | 30.29 | 35.72 | 5.71 |
| AB-max | 1224.06 | 195.85 | 217.41 | 33.45 | 36.75 | 5.88 |
| AB-trace | **1244.34** | **199.09** | **239.09** | **36.78** | **52.84** | **8.46** |

Table 5: ESS on 100d Sparse Gaussian

| Distribution | $L = 2$ | | $L = 10$ | | $L = 50$ | |
|---|---|---|---|---|---|---|
| Sampler | $\mathrm{ESS}_n$ | $\mathrm{ESS}_t$ | $\mathrm{ESS}_n$ | $\mathrm{ESS}_t$ | $\mathrm{ESS}_n$ | $\mathrm{ESS}_t$ |
| Gibbs | 3.80 | 15.21 | 4.20 | 16.81 | 4.42 | 17.68 |
| RW | 11.80 | 3.37 | 8.67 | 2.48 | 9.99 | 2.85 |
| LB | 147.91 | 28.17 | 71.23 | 13.57 | 32.19 | 6.13 |
| AB-1st | 247.59 | 47.16 | 79.88 | 13.31 | 36.62 | 6.98 |
| AB-shift | 545.06 | 87.21 | 154.00 | 24.64 | 45.19 | 7.23 |
| AB-max | 680.15 | 108.82 | 474.75 | 75.96 | 132.88 | 21.26 |
| AB-trace | **916.76** | **146.68** | **1227.94** | **196.47** | **1417.57** | **226.81** |

### B.4 BAYESIAN LOGISTIC REGRESSION

We consider the logistic regression model $Y \sim \text{Bernoulli}(\text{sigmoid}(X\beta))$, with $Y \in \{0,1\}^{50}, X \in \mathbb{R}^{50 \times 100}, \beta \in \{0,1\}^{100}$. We first generate $X \in \mathbb{R}^{50 \times 100}$. Each row $X_i$ is a realization of the normal distribution $\mathcal{N}(0, 0.25 \Lambda \Sigma \Lambda)$, where $\Sigma_{ii} = 1.25$, $\sigma_{ij} = 0.25$, and $\Lambda_{ii} = \exp(-0.25 + (i-1)/99)$. Then, we set the ground truth $\beta$ that $\beta_i = 1$ for $i = 1, 2, ..., 7$ and $\beta_i = 0$ for $i = 8, 9, ..., 100$. Then, we get the logits $v = X\beta$. Then, we sample $Y_i \sim \text{Bernoulli}(\sigma(v_i))$ for $i = 1, ..., 50$, where $\sigma(t) = 1/(1 + \exp(-t))$. Our target distribution is the posterior of $\beta$ and the log probability function is:

$$f(\beta) = -\sum_{i=1}^{m} y_i \log\left(1 + \exp(-\sigma_i)\right) + (1 - y_i) \log\left(1 + \exp(\sigma_i)\right), \quad \sigma_i = \sum_{j=1}^{d} X_{ij} \beta_j \quad (43)$$

### B.5 QUARTIC MIXTURE MODEL

Following Rhodes & Gutmann (2022), we consider quartic mixture model, where the log likelihood function can be written as:

$$f(x) = \log\left(\sum_{k=1}^{K} \exp(-\text{poly}_k^4(x))\right) \quad (44)$$

where $\text{poly}_k^4$ is multivariate polynomial with degree 4 generated in the following way. For component $k = 1, ..., 50$, the associated bias $b_k = \frac{k-1}{49}\mathbf{1} \in \mathbb{R}^{50}$. Denote $s \in \mathbb{R}^{20}$ such that $s_i = 20 * \exp(-0.5 + (i-1)/19)$ for $i = 1, ..., 20$. Then, we have the vector

$$v_k = (x - b_k)/s_i \in \mathbb{R}^{20 \times 50} \quad (45)$$

We also generate a rotation matrix $P \in \mathbb{R}^{20}$ shared by all components and we have :

$$u_k = P v_k \in \mathbb{R}^{20 \times 50} \quad (46)$$

Then, the polynomial is defined as

$$\text{poly}_k^4(x) = \sum_{d=1}^{50} (u_d + 1)(u_d^2 + 0.5)(u_d - 2) \quad (47)$$

### B.6 DEEP EBM

**Model training**. In the main text we compare the sampling efficiency of different samplers using the trained deep EBMs. Here we provide more details on obtaining these pretrained EBMs.

We follow the existing works to parameterize the EBMs using ResNets, where it is trained using persistent contrastive divergence (Tieleman, 2008) framework. Specifically we follow Grathwohl et al. (2021); Sun et al. (2021) to maintain a buffer of multiple MCMC chains. We use the sampler proposed in Sun et al. (2021) and run 60 steps to obtain samples from the current model per each gradient update. We retain the model after 50,000 steps of training. The models are all reasonable and can produce realistic binary images as the ground truth data.

**Estimating $W$**. Since we compare the mixing time on real distributions, we can not use AB sampler as Algorithm 1 which always use $W = D = 0$ during burn-in stage. Instead, we directly use the true data from the datasets to estimate the variance $\text{var}_i$ for each variable. Then we set $W$ as diagonal matrix with $W_{ii} = 1/(1 + \text{var}_i)$. In this case, the proposal distribution in equation 11 is naturally factorized and we don't need to use Gaussian integral trick.

**Adaptive Tuning**. The adaptive Algorithm 3 tuning $(\sigma, \alpha)$ based on average jump distance, which could be very unstable during the mixing stage. Hence, we simply apply a grid search of the configurations of $(\sigma, \alpha)$, and report the best one in Figure 4.

