# OpenReview forum: "Any-scale Balanced Samplers for Discrete Space"
_ICLR.cc/2023/Conference — ICLR 2023 poster_

### Official Review · Reviewer_XxHt · 2022-10-18

**Confidence:** 4
**Correctness:** 3
**Technical Novelty And Significance:** 2
**Empirical Novelty And Significance:** 2
**Recommendation:** 5

**Clarity, Quality, Novelty And Reproducibility:**

The paper is clearly writtern. However, there is a lack of originality as the main idea seems to be explored in a previous paper by Rhodes and Gutmann, 2022.

**Strength And Weaknesses:**

Strength:

1. The paper is clear and well written.
2. non-local proposals are helpful for efficient sampling of discrete distributons and the paper introduces several techniques to deal with the associate problems.


Weaknesses:

1. The preconditioning (using $D$ instead of $\sigma^2 I$) seems to be a useful trick. The same trick can be applied to other baseline method (for example, DLP can also use anisotropic diagonal as well), additional comparison to these variants would better illustrate the merits of the second order quadratic approximation.

2. Lack of a comparison to the precondition version PAVG of Rhodes and Gutmann, 2022.

**Summary Of The Paper:**

This paper proposed an any-scale balanced proposal for discrete distribution. Motivated by recent findings that non-local proposals improve sampling efficient, the proposed method closes the gaps of the choice of weight function and the accuracy of first order gradient approximation. Comparison to several baseline method demonstrate the efficiency of the proposed method.

**Summary Of The Review:**

The paper provides useful tricks for hyperparameter choice but lacks originality. Also, a very important baseline method that is highly related to the proposed method is missing (PVAG in Rhodes and Gutmann, 2022).

---

> ### Author Response · Authors · 2022-11-18
> **Response to Reviewer XxHt**
>
> We thank the reviewer for providing comments, and we want to kindly point out the potential misunderstandings in our response below.
>
> ### "...Using D-preconditioning in locally balanced proposals..."
>
> Using D-preconditioning has limited benefit with weight function $g(t) = t^\frac{1}{2}$. To demonstrate this, we conduct extra experiments for DLP-trace, which is DLP [4] with anisotropic diagonal $D$ as AB-trace in kernel $K_\sigma(z) = \exp(- z^\top D z / \sigma)$. We report the results in our revised version in table 1 and figure 2, 3. For convenience we report $\text{ESS}_t$ for DLP-trace, DLP, AB-1st, and AB-trace here.
> | Methods | Grid Ising | BA-4 Ising | Rotation Gaussian | Sparse Gaussian | BLR | QMM|
> |---|---|---|---|---|---|---|
> | DLP | 0.49 | 20.89 | 15.69 | 13.57 | 35.60 | 12.25 |
> | DLP-trace | 0.47 | 24.90 | 15.75 | 27.21 | 40.69 | 12.58 |
> | AB-1st | 0.67 | 35.89 | 16.50 | 13.31 | 51.44 | 32.11|
> | AB-trace | 1.13 | 110.90 | 36.78 | 196.47| 57.68 | 42.28 |
>
> It can be seen that DLP-trace performs slightly better than DLP on most of the target distributions as we expected. More interestingly, DLP-trace is consistently less efficient than AB-1st, except for Sparse Gasussian, which has an extremely large condition number. AB-1st uses an isotropic kernel, and all the improvements in efficiency are obtained from tuning $\alpha$. This observation highlights one of our main contributions of any-scale balanced sampler, where one needs to carefully balance the weight function with the scales in proposal distributions.
>
>
> ### "...Lack of comparison with PAVG..."
>
> We kindly point out that we already had such a comparison in the *original* submission, where we denote it as AB-shift and described the equivalence in section 5.1, the last two sentences in paragraph “samplers”. We admit that one might miss this if one didn’t read through the text carefully, so we make it more explicit in our revised version by replacing the name “AB-shift” with “AB-shift (PAVG)”.
>
> ### "...Useful tricks for hyperparameter choice but lacks originality..."
>
> While both the PVAG and our work leveraged the well known Gaussian Integral trick, there are substantial differences in technical contributions, formulations and experimental results. We have already *appropriately cited and compared* with PVAG in our originally submitted draft, and we reiterate the key differences below:
>
> - Before this work, the importance of $\alpha$ in $g(t) = t^\alpha$ had not been identified. People had previously only used a locally balanced function like $g(t) = t^\frac{1}{2}$, following [1] which proves that locally balanced functions are asymptotically optimal for a local proposal, but such proposals were being applied outside the scope of the main theoretical support in performing global scope sampling. We believe that this paper’s observation that fixes this gap in non-local proposals is conceptually important. Meanwhile, empirically, tuning $\alpha$ brings substantial improvements in sampling efficiency.
>
> - Although PAVG is a concurrent work using the same well-known Gaussian integral trick, the proposed method is significantly different in how this trick is leveraged. Actually, there is a long history of using the Gaussian integral trick in sampling from discrete space [2, 3], and we did not claim credit for that. As far as we know, our work is the first to select the diagonal matrix $D$ via minimizing its trace. Empirically, AB-trace *significantly outperforms* other samplers based on the Gaussian integral trick, including AB-max and *PAVG* (denoted as AB-shift in our initial draft). Also, we formulate the problem of choosing $D$ as a semidefinite program, which allows for $D$ to be obtained for problems with medium size.
>
>
> We kindly request that the reviewer re-evaluate the contribution of this paper, given that we have already compared in detail with PVAG. We also look forward to answering further questions if any.
>
> **References**
>
> >[1] Benjamin Rhodes and Michael Gutmann. Enhanced gradient-based MCMC in discrete spaces. arXiv preprint arXiv:2208.00040, 2022.\
> [2] James Martens and Ilya Sutskever. Parallelizable sampling of Markov random fields. In Proceedings of the Thirteenth International Conference on Artificial Intelligence and Statistics, pp. 517–524. JMLR Workshop and Conference Proceedings, 2010.\
> [3] Yichuan Zhang, Zoubin Ghahramani, Amos J Storkey, and Charles Sutton. Continuous relaxations for discrete Hamiltonian Monte Carlo. Advances in Neural Information Processing Systems, 25, 2012\
> [4] Ruqi Zhang, Xingchao Liu, and Qiang Liu. A Langevin-like sampler for discrete distributions. In International Conference on Machine Learning, pp. 26375–26396. PMLR, 2022.

---

> > ### Comment · Reviewer_XxHt · 2022-11-22
> > **Thanks for clarification!**
> >
> > Thanks for the clarification and I have updated my score. While I am still concerned about the orginality of the work, the technical contribution is good. Below are some additonal comments.
> >
> > Thanks for the new DLP-trace result. Is the D retrained or borrowed from AB-trace directly?
> >
> > For the AB-shift experiment now called PAVG, are $\alpha$ and $\sigma$ also tunned or fixed? Note that in PAVG, $\alpha=1$ which is not locally balanced.

---

> > > ### Author Response · Authors · 2022-11-23
> > > **Follow Up**
> > >
> > > We thank the reviewer for carefully reading our response, and we really appreciate your effort in actively responding with further comments. We list our answers here, and will add them to our revision.
> > >
> > > ### “Is the D retrained or borrowed from AB-trace directly?”
> > >
> > > *In short:* it is re-trained rather than borrowed from AB-trace.
> > >
> > > *In details:* The diagonal matrix D for DLP-trace is obtained in the following way:
> > > - During burn-in period, we run DLP with tuning average acceptance rate being 0.574 to collect samples.
> > > - After burn-in period, we used the collected samples to solve the diagonal matrix $D_\text{trace}$ via minimizing the trace as in Equation (19).
> > > - During the mixed period, we use the matrix $D = D_\text{trace} + cI$ for DLP-trace to collect samples. The reason to introduce $cI$ is that we find that several diagonal values of $D_\text{trace}$ equal to $0$, which is not a desirable property of a kernel function. Hence, we also tune the value of $c$ to maximize the efficiency of DLP-trace.
> > >
> > > ### “For the AB-shift experiment, are $\alpha$ and $\sigma$ also tuned or fixed?”
> > >
> > > We leverage the same proposed automated tuning for the hyperparameters $(\sigma, \alpha)$ in AB-shift, as we found it always yields better results than the fixed version (i.e., the very-vanilla PAVG, see below for justifications). Instead of confounding tuning together with estimation of D, we focused on the comparison of different estimators of D in the experiments and used the same tuning mechanism for all the methods.
> > >
> > > For quadratic models like Ising and Gaussian, AB-shift is equivalent to PAVG, since the optimal $\alpha = 1$. For the Bayesian logistic regression (BLR) and the quartic mixture model (QMM), this would make a difference.
> > >
> > > We follow the reviewer’s suggestion to evaluate the PAVG with fixed hyperparameters. Since the original paper [1] of PAVG does not provide a criteria about how to choose $\sigma$ (which is denoted as $\epsilon$ in [1]), we enumerate different choices of $\sigma$. Specifically, we manually sweep over $\sigma=X$ and denote it as PAVG-X, and report together in the following table (in terms of $\text{ESS}_n$).
> > >
> > > | Sampler | PAVG-1| PAVG-4 | PAVG-16 | PAVG-64 | PAVG-256 | PAVG-1024 | AB-shift (PAVG + auto-$\sigma, \alpha$)|
> > > |---|---|---|---|---|---|---|---|
> > > | BLR | 159.72 |  241.20 | 257.42 |  260.68 | 260.91 | 260.75 |  265.89 |
> > >
> > > | Sampler | PAVG-1| PAVG-10 | PAVG-100 | PAVG-1e3 | PAVG-1e4 | PAVG-1e5 | AB-shift (PAVG + auto-$\sigma, \alpha$)|
> > > |---|---|---|---|---|---|---|---|
> > > | QMM | 6. 88 | 29.74 | 171.26 | 270.35|  273.48 | 274.77 |  327.88 |
> > >
> > > We can see that the vanilla PAVG needs careful tuning of the hyperparameters, and the auto-tuned version (the AB-shift) has better ESS compared to PAVG. Also, the advantage of AB-shift is more significant in QMM, as the optimal $\alpha=0.92$ in QMM is far from $1$ compared to the optimal $\alpha=0.96$ in BLR.
> > > In summary, both the auto-tuning of hyperparameters and the estimation of D are critical. In our initial draft we enabled auto-tuning for all the methods and focused on demonstrating the advantage of our proposed trace-based estimation method. We will also include the above discussion in our revisions.
> > >
> > > *References*
> > >
> > > >[1] Benjamin Rhodes and Michael Gutmann. Enhanced gradient-based MCMC in discrete spaces. arXiv preprint arXiv:2208.00040, 2022.

---

### Official Review · Reviewer_XywG · 2022-10-22

**Confidence:** 4
**Correctness:** 4
**Technical Novelty And Significance:** 2
**Empirical Novelty And Significance:** 3
**Recommendation:** 6

**Clarity, Quality, Novelty And Reproducibility:**

The authors clearly state their goals, show that their proposal fulfils their claims and are not shy of highlighting the limitations of their proposal. The paper is in my opinion clear and of very good quality. The results are new, thus the paper ticks the novelty criterion. I believe that enough details are given in the paper and the supplementary material to also tick the reproducibility criterion.

There are some minor typos and errors:
- page 2, paragraph above section 2, typo neumerical
- page 4, paragraph above section 3.2, typo hyperparamter
- page 5, paragraph beneath eq. 15, typo should gives
- page 5, paragraph "A common approach ...", shouldn't j=2,...,d instead of j=1,...,d ?
- page 7, paragraph "In this experiment ...", last sentence, it is said that more results are given in Table 2 and Table 3, however, it should be mentioned that they are to be found in the appendix since on the next page we have Table 2 and Table 3 which don't contain results for the Ising model

**Strength And Weaknesses:**

The proposed algorithm is clearly explained, I personally appreciated the step by step manner in which the algorithm is presented. In terms of performances, the algorithm offers notable improvements over its locally balanced counterparts. Results are interpreted and the limitations of the algorithm are clearly stated.

The authors highlighted rather well themselves the limitations of the proposed algorithm. I would actually count this as a strength, not a weakness. Personally, I don't see any other weaknesses.

**Summary Of The Paper:**

The paper provides a non-local extension of the locally balanced samplers for improved sampling efficiency. The authors start by providing an analysis of existing approaches to improve the sampling efficiency of locally balanced samplers. They then introduce the any-scale balanced sampler in a step by step manner. Numerical results showing the effectiveness of the proposed algorithm conclude the paper.

**Summary Of The Review:**

The paper proposes an any-scale balanced sampler, a non-local extension of locally balanced samplers. The proposed algorithm is clearly presented, numerical results that show the advantages of the proposed sampler are presented and interpreted. Limitations are clearly discussed. Good paper.

---

> ### Author Response · Authors · 2022-11-18
> **Response to Reviewer XywG**
>
> We thank the reviewer for recognizing our work. We also thank the reviewer for the careful reading and pointing out our typos and errors in presentation. We have fixed them in the revision.

---

> > ### Comment · Reviewer_XywG · 2022-11-19
> > **Read the authors rebuttal**
> >
> > I read the rebuttal and I would like to thank the authors for the effort in answering the questions raised during the review phase. My impression is still that the paper is worth accepting.

---

> > > ### Author Response · Authors · 2022-12-06
> > > **kindly asking for an explanation of the score changes**
> > >
> > > Dear reviewer,
> > >
> > > Thank you again for your effort during the review process, and your recognition to our work!
> > >
> > > We noticed that your rating has changed from 8 to 6 on Nov 27, one week after the rebuttal. This is a bit unexpected as it seems there's no new concern raised.
> > >
> > > We want to kindly check with you on this, in case there might be some unexpected changes. Or if there are other explicit reasons that we are not aware of, we want to follow up with you and see if we can have the chance to address.
> > >
> > > In any case, we fully respect your decision and we look forward to learning from your thoughts!
> > >
> > > Best,
> > >
> > > Authors

---

### Official Review · Reviewer_Ce4Q · 2022-10-24

**Confidence:** 5
**Correctness:** 4
**Technical Novelty And Significance:** 1
**Empirical Novelty And Significance:** 3
**Recommendation:** 6

**Clarity, Quality, Novelty And Reproducibility:**

## Clarity:
the article is clearly written and very easy to follow.

## Quality:
The proposed method is convincing and natural. It appears to empirically work extremely well.

## Novelty:
Relatively weak: the tuning part of $\alpha,\sigma$ is entirely straightforward. The tuning of $W$ is interesting. The use of the Gaussian trick is also very standard.

## Reproducibility:
Enough detail to reproduce everything.

**Details Of Ethics Concerns:**

no concern.

**Strength And Weaknesses:**

## Strengths
The paper reads well and the proposed approached is very natural and convincing. Furthermore, the simulations are quite well executed.

## Weakness
I do not see any particular weakness.
1. Indeed, the proposed methods is extremely natural after one has read the recent paper [1] but the article seems (?) to have been written at the same time.
2. I am not really sure we need the discussion at the bottom of page 3 (eg. Equations 5/6/7) since this does not seem to add much to the text. But it certainly does not hurt.

### Very Minor comments
1. The authors seem to parametrize Gaussian distributions with mean and standard-deviation $N(\mu, \sigma)$ instead of mean and variance $N(\mu, \sigma^2)$, which is a bit awkward (is it common in some communities?).
2. Should it be $K_\sigma$ instead of $H_\sigma$ in Equation 5. Also, $k_\sigma$ is used of $K_\sigma$ on page 2.

### References
1. Rhodes B, Gutmann M. Enhanced gradient-based MCMC in discrete spaces. arXiv preprint arXiv:2208.00040. 2022 Jul 29.

**Summary Of The Paper:**

The locally-balanced approach of Zanella has recently proven to be useful for MCMC sampling in discrete state spaces. The article proposes two main innovations when compared to recent articles building on the approach of Zanella:
1. tuning of the "balancing function" $g(t) = t^{\alpha}$ and length-scale $\sigma$ of the kernel $K_\sigma$. The tuning of the parameters $(\alpha, \sigma)$ is done quite naturally by attempting the maximise the "Expected Squared Jumping Distance" during an adaptation phase.
2. take second order effect by choosing a global "preconditioning" matrix such that $f(y) \approx f(x) + \nabla f(x)^\top (y-x) + \frac12 \langle (y-x), W (y-x) \rangle$. The choice of the global matrix $W$ is done during an adaptation phase by minimising $\|W(y-x) - (\nabla f(y) - \nabla f(x))\|^2$, which is quite cheap computationally to implement.

Using a second order approximation of the log target leads to sampling from a Ising-like distribution. As has already been discussed in other article [1], this can be approached by using the standard Gaussian integral trick (i.e. Hubbard-Stratonovich transformation). In order to implement this approach, a diagonal matrix $D$ such that $W+D$ is positive definite needs to be chosen. The authors propose to choose $D$ so that $W+D$ as "isotropic" as possible, which is a sensible and standard strategy that appears to work well in this case.


### References
1. Rhodes B, Gutmann M. Enhanced gradient-based MCMC in discrete spaces. arXiv preprint arXiv:2208.00040. 2022 Jul 29.

**Summary Of The Review:**

The article proposes an extension of the work of Zanella to take second order effect into account (i.e. preconditioning) when sampling from discrete distributions, which is very important in practical scenarios (i.e. preconditioning can lead to very large improvements in sampling efficiency). Even if the approach is relatively straightforward, it works well and the numerical simulations are convincing.

---

> ### Author Response · Authors · 2022-11-18
> **Response to Reviewer Ce4Q**
>
> We would like to thank the reviewer for their valuable feedback! Please see our response below.
>
> ### Novelty and Contribution of selecting $\alpha, \sigma$
>
> The main novelty is that we observe the gap of using locally balanced weight functions in nonlocal proposals. The recently proposed samplers all use the locally balanced function (e.g.,  $g(t) = t^\frac{1}{2}$), following the pioneering work of Zanella, who proved that the locally balanced function is asymptotically optimal for a local proposal. However many of these recently proposed samplers are actually doing sampling in much larger scopes, where the weight function is no longer optimal. We are the first to realize this problem, and defined a family of weight functions $g(t) = t^\alpha$ where $\alpha$ can be changed with the scale $\sigma$.
>
> It might seem intuitive to optimize $\alpha$ once the problem has been identified and the tuning objective defined. But identifying and defining the problem was nontrivial and this had not yet appeared in the literature, to the best of our knowledge. Moreover, there are still non-trivial technical difficulties involved in the tuning. The defined family $g(t) = t^\alpha$ may not be the only family that can achieve the any-scale balanced property, proving its uniqueness or finding other families of functions is yet to be solved; and even in this limited case, choosing the optimal $\alpha$ for an arbitrary family of scaling kernels could still be difficult. We hope to investigate these technical issues further in future works as well.
>
>
> ### Novelty of selecting the diagonal matrix $D$
>
> We agree with the reviewer that the Gaussian integral trick is a well-known technique as we cited in our paper, and we did not claim credit for that. In fact, the efficient way for selecting $D$ is our novel contribution.
> Specifically, we first proposed a novel trace minimization objective, where the corresponding AB-trace sampler empirically outperforms other “isotropic” or “anisotropic” methods, e.g. AB-shift (or equivalently the PVAG [1]) that uses $D=\lambda I$ and AB-max that minimizes the maximum eigenvalue of $(W+D)$. Second, we formulate the problem of choosing $D$ as a semidefinite programming (SDP) problem, which allows efficient solving for problems with medium size.
>
> ### Concurrent work [1]
>
> Indeed our work and [1] were developed independently. However, [1] appeared on arXiv earlier, while we chose to submit to ICLR. Nevertheless, we have properly cited and experimentally compared with [1] in our paper.
>
> We would like to re-emphasize that despite the overlaps in using the well-known Gaussian integral trick, our two main contributions—as pointed out by the reviewer on 1) balancing the weight function with the scale, and 2) selecting diagonal matrix D via SDP— have not been discussed in [1].
>
> Moreover, empirically the resulting AB-trace sampler has demonstrated considerable advantages in efficiency, compared to [1] (denoted as AB-shift in our initial draft).
>
> ### Discussion at the bottom of page 3
>
> The motivation to give this discussion is to provide a concrete and analytical example to show that locally balanced function is **not** optimal for non-local proposals.
>
>
> ### “Typo: Should it be $K_σ$ instead of $H_σ$ in Equation 5. Also, $k_σ$ is used of $K_σ$ on page 2.”
>
> We thank the reviewer for pointing out the typos. We have fixed them in the revised version.
>
>
> **References**
> >[1] Benjamin Rhodes and Michael Gutmann. Enhanced gradient-based MCMC in discrete spaces. arXiv preprint arXiv:2208.00040, 2022.

---

> > ### Comment · Reviewer_Ce4Q · 2022-11-21
> > **Thanks.**
> >
> > Thank you for your replies. I agree with your comments (and am still finding your contribution interesting!).

---

### Decision · Program_Chairs · 2023-01-20

**Decision:**

Accept: poster

**Justification For Why Not Higher Score:**

The reviewers concluded that paper had interesting ideas, though neither the novelty of the algorithmic improvements, nor the experimental gains merited a recommendation higher than a poster.

**Justification For Why Not Lower Score:**

The ideas of the paper improve a line of work that is quite successful at designing discrete space samplers and deserves a place at the conference.

**Metareview: Summary, Strengths And Weaknesses:**

The paper considers two improvements to a recent, locally-balanced sampler approach of Zanella used in the context of discrete state spaces. Precisely, the authors propose a natural to tune $\alpha$ in the balancing function $g(t) = t^{\alpha}$ and the variance $\sigma$ in the length-scale of the kernel $K$. They also propose a way to fit a global "preconditioning" matrix for a second-order term in the Taylor expansion of $f$, which is relatively cheap to compute and intended to aid in the case where the steps are large so that the first-order expansion is inaccurate.

The reviewers all thought that the ideas of the paper were clean, easy to implement, and described well. The experiments were also encouraging and executed well. The main concern was the amount of overlap with a (presumably concurrent) work by Rhodes and Gutman, but the decision was made that due to the fact that the works were simultaneous, and comparison to the paper was provided, this is ok.

**Note From Pc:**

if the above contains the word "oral" or "spotlight" please see: "oral" presentation means -> notable-top-5% and "spotlight" means -> notable-top-25%. As stated in our emails, we are disassociating presentation type from AC recommendations